# Glibenclamide-Loaded Engineered Nanovectors (GNVs) Modulate Autophagy and NLRP3-Inflammasome Activation

**DOI:** 10.3390/ph16121725

**Published:** 2023-12-13

**Authors:** Marina Saresella, Chiara Paola Zoia, Francesca La Rosa, Chiara Bazzini, Gessica Sala, Erica Grassenis, Ivana Marventano, Ambra Hernis, Federica Piancone, Elisa Conti, Silvia Sesana, Francesca Re, Pierfausto Seneci, Carlo Ferrarese, Mario Clerici

**Affiliations:** 1IRCCS Fondazione Don Carlo Gnocchi, 20147 Milan, Italy; msaresella@dongnocchi.it (M.S.); imarventano@dongnocchi.it (I.M.); ahernis@dongnocchi.it (A.H.); fpiancone@dongnocchi.it (F.P.); mario.clerici@unimi.it (M.C.); 2Neurobiology Laboratory, School of Medicine and Surgery, University of Study of Milano-Bicocca, 20900 Monza, Italy; chiarapaola.zoia@unimib.it (C.P.Z.); chiara.bazzini@unimib.it (C.B.); gessica.sala@unimib.it (G.S.); e.grassenis@campus.unimib.it (E.G.); elisa.conti@unimib.it (E.C.); carlo.ferrarese@unimib.it (C.F.); 3Milan Center for Neuroscience, University of Study of Milano-Bicocca, 20126 Milano, Italy; 4BioNanoMedicine Center NANOMIB, School of Medicine and Surgery, University of Milano-Bicocca, 20126 Milan, Italy; mariasilvia.sesana@unimib.it (S.S.); francesca.re1@unimib.it (F.R.); 5Department of Chemistry, University of Milan, Via Golgi 19, 20133 Milan, Italy; pierfausto.seneci@unimi.it; 6Department of Neuroscience, IRCC Fondazione S. Gerardo dei Tintori, 20900 Monza, Italy; 7Department of Pathophysiology and Transplantation, University of Milan, 20122 Milan, Italy

**Keywords:** autophagy, Glibenclamide-loaded engineered NanoVectors (GNVs), NLRP3

## Abstract

Activation of the NLRP3 inflammasome in response to either exogenous (PAMPs) or endogenous (DAMPs) stimuli results in the production of IL-18, caspase-1 and IL-1β. These cytokines have a beneficial role in promoting inflammation, but an excessive activation of the inflammasome and the consequent constitutive inflammatory status plays a role in human pathologies, including Alzheimer’s disease (AD). Autophagic removal of NLRP3 inflammasome activators can reduce inflammasome activation and inflammation. Likewise, inflammasome signaling pathways regulate autophagy, allowing the development of inflammatory responses but preventing excessive and detrimental inflammation. Nanotechnology led to the development of liposome engineered nanovectors (NVs) that can load and carry drugs. We verified in an in vitro model of AD-associated inflammation the ability of Glibenclamide-loaded NVs (GNVs) to modulate the balance between inflammasome activation and autophagy. Human THP1dM cells were LPS-primed and oligomeric Aß-stimulated in the presence/absence of GNVs. IL-1β, IL-18 and activated caspase-1 production was evaluated by the Automated Immunoassay System (ELLA); ASC speck formation (a marker of NLRP3 activation) was analyzed by FlowSight Imaging flow-cytometer (AMNIS); the expression of autophagy targets was investigated by RT-PCR and Western blot (WB); and the modulation of autophagy-related up-stream signaling pathways and Tau phosphorylation were WB-quantified. Results showed that GNVs reduce activation of the NLRP3 inflammasome and prevent the Aß-induced phosphorylation of ERK, AKT, and p70S6 kinases, potentiating autophagic flux and counteracting Tau phosphorylation. These preliminary results support the investigation of GNVs as a possible novel strategy in disease and rehabilitation to reduce inflammasome-associated inflammation.

## 1. Introduction

Autophagy is a complex process that can be differentiated into three primary types of autophagy: microautophagy, macroautophagy, and chaperone-mediated autophagy (CMA). While each is morphologically distinct, all three culminate in the delivery of cargo to the lysosome for degradation and recycling [1]. In macroautophagy, the autophagic flux is a non-specific process which can be divided into four phases: induction, autophagosome formation, maturation or fusion with lysosomes, and degradation. Macroautophagy is physiologically inactivated by mTOR signaling and relies on the activity of two key proteins, Beclin-1 and microtubule-associated protein light chain 3 (LC3). These proteins allow the formation of autophagic lysosomes, where cargoes are degraded by lysosomal enzymes and degradation products are recycled [1]. p62 is an autophagy substrate that is used as a reporter of autophagy activity; notably, p62 is associated with neurofibrillary tangles (NFTs) and is believed to play an important role in Tau degradation [2]. CMA, on the other hand, is selective for a particular group of cytoplasmic proteins which are translocated across the lysosomal membrane in the absence of vacuole formation. CMA substrates contain the KFERQ (Lys-Phe-Glu-Arg-Gln) amino acid sequence, which is recognized by chaperones, including the 70 kDa heat shock protein (Hsc70). The cargo-Hsc70 complex is transported to the lysosomal membrane, where it interacts with the lysosome-associated membrane protein type 2A (lamp2A) cytosolic domain. Once the cargo-Hsc70 complex binds to LAMP2A, Hsc70 is released and lamp2A oligomerizes to form a pore. The cargo protein is first unfolded and then degraded [3].

CMA and/or macroautophagy can be impaired in a number of diseases, including Alzheimer’s disease (AD). In AD, the observations that (1) a C-terminal KFERQ motif is present in both Tau and Amyloid Precursor Proteins (APPs), (2) mutations in the Tau protein reduce CMA activity, and (3) the efficacy of CMA declines with age help explain the pathogenesis of disease-associated autophagy alterations [3]. AD is nevertheless the result of complex interactions between multiple pathological processes that include Aβ alterations, Tau phosphorylation, neurotransmitter dysregulation, oxidative stress, neuroinflammation, and alterations in autophagy. Neuroinflammation, in particular, modulates autophagy and autophagy modulates the activation of the NLRP3 inflammasome [4,5]. Modulation of autophagy depends as well on the status of other proteins. Extracellular signal-regulated kinase 1/2 (ERK1/2) and p38, which are members of the mitogen-activated protein kinase (MAPK) family, play a key role in regulation of autophagy. Several findings suggest that ERK1/2 are dysregulated in AD and suggest this to be an important factor in Aβ plaque formation, Tau phosphorylation, and neuroinflammation [6]. Alterations of phosphatidylinositol 3-kinase/AKT (PI3K/AKT) signaling, observed as well in AD, modulate autophagy and contribute to Tau phosphorylation, Aβ toxicity, and neuroinflammation [5,7,8]. Notably, the PI3K/AKT and ERK1/2 signaling pathways converge on the common downstream effector p70S6K, another modulator of autophagy whose activation is associated with Tau hyperphosphorylation at Ser262 and Ser214 [9,10].

Neuroinflammation in AD results from an increased production of proinflammatory cytokines by the microglia [11] and the migration of activated peripheral monocytes across the blood–brain barrier (BBB) [12,13], and it is correlated with a specific pattern of brain functional disconnection [11,12,13,14,15]. Activation of the NLRP3-inflammasome complex in microglia and in peripheral monocytes plays an important role in the production of the inflammatory cytokines IL-1β and IL-18 and could be envisioned as a therapeutic target [16,17,18,19]. In recent years, a large amount of data from cell experiments and animal models have confirmed that the activation of NLRP3 inflammasome can also affect the deposition and spread of Aβ. APP/PS1 mice, compared to NLRP3 and caspase-1 knockout AD model mice, have a significantly enhanced the ability of microglia to phagocytose Aβ and differentiate microglia into anti-inflammatory M2 type, which facilitates Aβ clearance [13].

Different compounds can reduce the activation of the NLRP3 inflammasome; Stavudine (DT4), a nucleoside reverse transcriptase inhibitor, has this effect and stimulates Aβ autophagy by macrophages [20]. Glibenclamide, (Gli) a second generation sulfonylurea with hypoglycemic action, can also down-regulate NLRP3 inflammasome activation and, as a consequence, might have a positive effect in reducing neuroinflammation [21,22]. The majority of drugs, including Gli, penetrate the BBB very poorly; to bypass this problem, liposome-based nanocarriers that can carry drugs across the BBB have been engineered [23]. These innovative NanoVectors (NVs) are multi-functionalized liposomes carrying mApoE to recognize and cross the BBB, and protease-sensitive peptides that drive drug release in the inflammatory niche where specific matrix metalloproteinases (MMP) are overexpressed.

To test the efficacy of GNVs to modulate NLRP3 inflammasome-mediated inflammation and autophagic processes, we tested their effect in an in vitro model of LPS + Aβ stimulated THP-1dM cells. Results showed that GNVs greatly reduce NLRP3 inflammasome activation, up-regulating macroautophagy as well as the molecular mechanisms that regulate Aβ cytotoxicity and Tau phosphorylation.

## 2. Results

### 2.1. GNVs Reduce Pyroptosis and Increase Mitochondrial Activity

Lactate dehydrogenase (LDH) is a stable cytoplasmic enzyme found in all cells and released into supernatants during pyroptosis; we measured LDH activity in supernatants collected by LPS + Aβ stimulated THP-1 dM cells in the presence/absence of GNVs as a proxy, to determine their anti-pyroptotic effect. Results showed that, compared to the values observed in cells cultured in medium alone, LDH activity was significantly increased when cells were LPS + Aβ-stimulated in the absence of GNVs (median: 0.9 Mu/mL), but it was completely downregulated upon addition of GNVs (0.2 mU/mL; *p* = 0.002) (Figure 1A). To evaluate the possible antioxidant role of GNVs, mitochondrial activity was analyzed next by MTT. Results showed that LPS + Aβ stimulation was harmful for mitochondria, reducing their activity (vs. medium alone: *p* < 0.01); this was completely prevented by GNVs (vs. LPS + Aβ *p* < 0.001) (Figure 1B). GNVs treatment did not modify mitochondrial activity compared to untreated cells (MED), and for this reason, GNVs are biocompatible.

### 2.2. GNVs Effect in ASC Speck Formation, NLRP3-ASC Colocalization and Downstream NLRP3-Inflammasome Activation in THP-1 dM

A hallmark of NLRP3 inflammasome activation is ASC speck, a micrometer-sized structure formed by the inflammasome adaptor protein ASC (apoptosis-associated speck-like protein containing a CARD), which consists of a pyrin domain (PYD) and a caspase recruitment domain (CARD). The oligomerization of ASC creates a multitude of potential caspase-1 activation sites, thus serving as a signal amplification mechanism for inflammasome-mediated cytokine production. Results showed that, as compared to cells cultured in medium alone (median: 0.3%), ASC speck-expressing cells were greatly augmented in LPS + Aβ-stimulated THP1dM cells in the absence of GNVs (median: 15%; *p* < 0.01); the addition of GNVs completely suppressed ASC-speck formation (median 0.2%) (absence/presence of GNVs *p* > 0.01) (Figure 2A). The percentage of positive cells for ASC-speck formation (differently to ASC-diffuse) was selected (Figure 2D); the NLRP3 production and ASC-speck colocalization were investigated next by Flowsight AMNIS analyses in LPS + Aβ-stimulated THP1dM cells in the absence of GNVs (Figure 2E,F). Results were shown as percentage (median:11%) of double positive cells for ASC-speck and NLRP3 in LPS + Aβ-stimulated THP1dM in the absence of GNVs (Figure 2G). The addition of GNVs suppressed ASC-speck, thus the percentage of double positive cells was naught. Representative images are provided in Figure 2B,C,E,F.

Not surprisingly, GNVs significantly reduced the production of NLRP3 activation-associated proinflammatory cytokines as well. Thus, as compared to the values observed in LPS + Aβ-stimulated THP1dM cells in the absence of GNVs, caspase 1 (*p* = 0.001), IL-1β (*p* = 0.004), and IL-18 (*p* = 0.02) production was significantly reduced by GNVs (Figure 3A–C).

### 2.3. GNVs’ Effect on Macroautophagy and CMA

To verify whether GNVs modulate the two main autophagic pathways involved in Aβ clearance, macroautophagy and CMA, gene and protein expressions of selected macroautophagy (beclin-1, LC3 and p62) and CMA (lamp2A and hsc70) markers were evaluated by qPCR and Western blot (WB), respectively, in LPS + Aβ-stimulated THP1 cells in the absence/presence of GNVs.

As shown in Figure 4A,B, LPS + Aβ stimulation of THP1 cells resulted in a significant increase in beclin-1 (*p* < 0.05), p62 (*p* < 0.05) and lamp2A (*p* < 0.01) mRNA. LC3 mRNA was increased as well, although not significantly, whereas no differences were seen in hsc70 mRNA. A significant increase in beclin-1 and p62 (*p* < 0.05) protein expression and a tendency to decrease LC3-II protein, the lipidated form of LC3 used as an autophagosome marker, were found in LPS + Aβ-stimulated cells (Figure 4C,E), whereas no effect was observed on lamp2A and hsc70 protein levels (Figure 4D,F). Exposure to GNVs significantly reduced LC3-II protein (*p* < 0.01), as well as beclin-1 mRNA and protein expression, but it did not modify p62, hsc70, and lamp2A mRNA and protein expression.

Collectively, these results suggest that LPS + Aβ stimulation increases the expression of autophagy-associated genes and proteins, although the observed mild reduction in LC3-II protein suggests that an initial induction of macroautophagy is ongoing in cells after LPS + Aβ exposure, likely as a protective cell response against the inflammatory/toxic stimuli. Importantly, the treatment with GNVs, which significantly reduce autophagosome accumulation (as indicated by the reduced LC3-II protein level), seems to indicate a potentiation of the macroautophagic flux, with no effect on the CMA pathway (as evidenced by the unchanged lamp2A protein level). Although in the presence of macroautophagy upregulation, a decrease in the substrate p62 is expected, our results show an increase in both the gene and protein p62 levels after LPS + Aβ stimulation, independently from the presence of GNVs.

### 2.4. GNVs Modulation of the MAPK (ERK and p38) and PI3/AKT-Pathways

The NLRP3 inflammasome can also regulate autophagy by modulating the expression of phosphatases and kinases. The phosphorylation status of ERK and p38 MAPkinase, PI3/AKT, and p70S6 Kinase, a downstream target of the ERK and AKT pathways, was analyzed by WB in LPS + Aβ-stimulated THP1 dM cells in the absence/presence of GNVs. Results showed that LPS + Aβ stimulation induces protein phosphorylation, significantly increasing p-ERK (Figure 5A, *p* < 0.05), p-AKT (Figure 5B, *p* < 0.001), and p-p38 (Figure 5C, *p* < 0.01). These data indicate that LPS + Aβ stimulation results in down-regulation of autophagy, due to (1) ERK and AKT phosphorylation which activates mTOR, and (2) mTOR activation inhibits autophagy. ERK and AKT protein phosphorylation was prevented by GNVs (Figure 5A,B, *p* < 0.01 and *p* < 0.05, respectively), resulting in mTOR deactivation and, finally, autophagy upregulation. In contrast with their effect on ERK and AKT phosphorylation, GNVs did not counteract, but rather increased p38 phosphorylation (Figure 5C, *p* < 0.01).

### 2.5. GNVs Modulation of p70S6K and Tau Protein

p70S6K activation induces the phosphorylation of the Tau protein at Ser262 and regulates its gene transcription. LPS + Aβ stimulation of THP1-dM cells significantly increased phosphorylation of both p70S6K (*p* < 0.001) and Tau protein at Ser262 (*p* < 0.01) (Figure 6A,B); this was reversed by GNVs (p70S6K *p* < 0.01; Tau *p* < 0.05). GNVs, thus, may also have a role in modulating Tau hyperphosphorylation, a pathological hallmark of AD.

## 3. Discussion

Inflammation plays an important pathogenic role in neurodegenerative diseases, including AD. Thus, the concentration of proinflammatory cytokines, including IL-1β and IL-18 is upregulated in the brain and cerebrospinal fluid (CSF) of AD patients [24,25]. This is driven both by an increased production of these cytokines by microglia and by their ability to penetrate the BBB, thereby promoting inflammation in the brain [26] and mediating crosstalk between the immune and the central nervous systems [11,14,27,28,29,30,31]. Notably, increasing evidence indicates that the activation of the NLRP3 inflammasome plays a pivotal role in driving neuroinflammation in neurodegenerative diseases [15,32,33,34,35,36,37].

Under normal physiological conditions, activation of the NLRP3 inflammasome contributes to pathogens’ elimination and tissue repair [38]. However, dysregulated or excessive NLRP3 inflammasome activation is described in a number of pathologic conditions, including AD, diabetes, gout, autoimmune diseases, and atherosclerosis [39,40]. Thus, as such, NLRP3 inflammasome is a promising therapeutic target in chronic inflammatory diseases [41], including AD. Indeed, given the lack of effective drugs in the therapy of neurodegenerative conditions and the likely role of the NLRP3 inflammasome in the pathogenesis and progression of AD, efforts should be made to develop effective therapeutic strategies targeting the NLRP3 inflammasome.

In recent decades, nanoparticles have been used to enhance the efficacy of drugs designed to target the CNS. Increased drug bioavailability, limited undesirable side effects, and an increased accumulation of neuroprotective agents in CNS compartments were obtained [42,43]. In particular, the surface modification of nanoparticles with peptides, small molecules, or antibodies able to cross the BBB and to recognize specific brain disease targets have been exploited to improve the CNS-targeted delivery of nano-formulations [44]. Nanovectors (NVs) could be loaded with bioactive lipids with anti-inflammatory properties. Here, as a proof of concept, we tested, using in vitro experiments, the efficacy of GNVs to dampen the activation of the NLRP3 inflammasome. Gli is a hypoglycemic agent that can inhibit NLRP3 mediated production of IL-1β by blocking P2X7 ion-channels; this compound also modulates TRPM4, a cation channel that mediates axonal and neuronal degeneration in Multiple Sclerosis (MS) [45]. Despite these anti-inflammatory effects, Gli failed to achieve therapeutic levels in the brain and CSF [46,47], and thus, its encapsulation within NVs could represent a valid therapeutic improvement.

Notably, THP-1 cell lines constitutively express NLRP3 [48]; thus, to better investigate NLRP3 activation or not after GNVs treatments NLRP3 colocalization with ASC, ASC Speck formation, and related cytokines production were evaluated. Results showed a significant reduction in ASC-speck formation as well as of caspase-1, IL-18 and IL-1β production by THP-1 dM cells that had been LPS + Aβ-stimulated in the presence of GNVs. LDH production, a proxy to measure pyroptosis, was also significantly reduced by GNVs. These data are in accordance with recent studies, demonstrating that Gli ameliorates cognitive impairments in animal models of AD, decreasing pro-inflammatory cytokines production and pyroptosis in the hippocampus [49,50]. We have previously shown that the NLRP3 inflammasome is activated in monocytes of AD and in Aβ-stimulated THP-1 dM cell lines, resulting in increased IL-1β and IL-18 production and inefficient Aβ-phagocytosis. Both drugs [19], which inhibited NLRP3 activation, greatly reduced cytokine production but did not reactivate Aβ-phagocytosis, rather stimulating autophagic pathways. Because autophagy has a negative regulatory effect on NLRP3 inflammasome activation, we analyzed the ability of GVNs to modulate the NLRP3-autophagy interaction. Two main autophagic pathways are involved in Aβ clearance, macroautophagy, and chaperone-mediated autophagy (CMA); herein, we analyzed both pathways by studying gene expression and protein production of macroautophagy (beclin-1, LC3 and p62) and CMA (lamp2A and hsc70) markers.

Collectively, results indicate that the stimulation of THP-1 dM cells with LPS–Aβ significantly activates the transcription of macroautophagy-related genes (increased mRNA levels of beclin-1, p62, and LC3), which results in increased beclin-1 and p62 protein levels. Cell stimulation in the presence of GNVs was associated with a strongly reduced number of autophagosomes (lower LC3-II protein levels with consequent increased gene transcription as a feedback regulatory mechanism), which is consistent with a potentiation of the autophagic flux. Unexpectedly, protein levels of the macroautophagy substrate p62 were not reduced by GNVs, possibly because of a high gene transcription due to LPS + Aβ stimulation, independently of GNVs. These results are not surprising considering the multiple roles exerted by p62 in pathways activated in response to inflammatory stimuli and agree with published studies demonstrating the involvement of p62 not only in autophagy degradation, but also in inflammatory pathways under stress conditions [50,51,52]. As a consequence, since the exposure to GNVs was unable to markedly modify the increase in mRNA and protein levels of p62 induced by LPS + Aβ, we could speculate that p62 in these experimental conditions might be mainly engaged in the above-cited pathways rather than in the macroautophagy process.

Notably, while GNVs modulated macroautophagy, they did not have any impact on CMA, as indicated by the lack of any effect on gene and protein expressions of the CMA markers lamp2A and hsc70. It should be noted that a marked increase in lamp2A mRNA, which did not result in increased protein levels, was found in LPS + Aβ-stimulated cells, accordingly to the known increase in mRNA lamp2A levels under oxidative stress or other stressful conditions [53].

A putative effect of GNVs on the ERK and AKT pathways and their downstream effector p70S6K was also investigated. p70S6K is involved in the regulation of cell-cycle, in autophagy signaling, and in the phosphorylation of the Tau protein. In particular, p70S6K can phosphorylate Tau at the Ser262, Ser214, and Thr212 residues, which leads to Tau release from microtubules, resulting in microtubule disruption [10]. In recent decades, the concepts of Aβ cascade and Tau hyperphosphorylation have been central in AD drug development. However, Aβ- and Tau-directed agents have had very limited clinical success.

Herein, intracellular cell signaling was analyzed. Results showed that GNVs hamper ERK1/2, AKT, and p70S6K phosphorylation, possibly upregulating mTOR-dependent autophagy to reduce LPS + Aβ toxicity and Tau phosphorylation. In fact, LPS + Aβ-induced cytotoxicity upregulated ERK1/2 and AKT phosphorylation, activating mTOR and blocking autophagy. On the other hand, GNVs prevented the mTOR-dependent inhibition of autophagy. Furthermore, GNVs avoided the LPS + Aβ-dependent hyperphosphorylation of Tau. Taken together, our findings support the hypothesis that the NLRP3 inflammasome is a potential therapeutic target for AD and suggest GNVs as an effective treatment to prevent cytotoxicity and disease-related oxidative stress.

## 4. Materials and Methods

### 4.1. Cell Cultures

THP-1 (human acute monocytic leukemia cell line) (IZSLER, Istituto Zooprofilattico Sperimentale della Lombardia e dell’Emilia Romagna, IT) were cultured in RPMI 1640 (EuroClone, Pero, Italy) supplemented with 10% heat-inactivated fetal bovine serum (FBS) (EuroClone), 1% glutamine (EuroClone), 1% penicillin-streptomycin (EuroClone) at 37 °C in 5% CO_2_. Culture medium was changed every two days. Cells were seeded in 6-well plates at a density of 1 × 10^6^ cells/well; 10 ng/mL phorbol 12-myristate 13-acetate (PMA, Sigma-Aldrich, St. Louis, MO, USA) was added to the wells for 12 h at 37 °C in 5% CO_2_ to achieve differentiation into macrophages (THP-1 derived Macrophage_THP-1dM). Cells were cultured in medium alone (MED) or were primed for 2 h with LPS (1 μg/mL) and then activated with 2.5 µM Aβ (Phoenix Pharmaceuticals) for 22 h in absence/presence of 10µM GNVs (preparation and characterization by NANOMIB, as previously reported in [21]).

After incubation, THP-1dM cells were washed with PBS, harvested by adding 60 μL/well of Trypsin-EDTA solution (Sigma-Aldrich) for 5 min at room temperature (RT) and centrifuged for 10 min at 1500× *g*. Cytotoxicity was assessed by quantifying lactate dehydrogenase (LDH) release. LDH activity in supernatants was measured by the rate of reduction of NAD^+^ to NADH after addition of the LDH reaction mix according to the kit protocol (ab102526, Abcam), and it was calculated according to the color produced at 450 nm at two time points. The collected pellet was used for RNA extraction and for imaging flow analysis (see below). Supernatants of cell culture were frozen at −80 °C.

### 4.2. Mitochondrial Activity and Cell Viability (MTT) Assay

The MTT assay involves the conversion of the water-soluble yellow dye MTT [3-(4,5-dimethylthiazol-2-yl)-2,5-diphenyltetrazolium bromide] to an insoluble purple formazan by the action of mitochondrial reductase. Formazan is then solubilized, and the concentration determined by optical density at 570 nm. The result is a sensitive assay with excellent linearity up to ∼10^6^ cells per well.; small changes in metabolic activity can generate large changes in MTT, allowing one to detect cell stress upon exposure to a toxic agent in the absence of direct cell death. Herein, we tested GNVs biocompatibility and their ability to prevent LPS + Aβ toxic effects. THP-1dM cells were treated with 3-4,5-dimethylthiazol-2-yl)-2,5-diphenyl-2H-tetrazolium bromide (Sigma-Aldrich) 0.5 mg/mL in RPMI 1640 medium and incubated 45 min at 37 °C in an atmosphere of 5% CO_2_ in air. Cells then have been dissolved with DMSO, and absorbance was quantified at 570 nm using a multi-mode microplate reader (FLUOstar Omega, BMG LABTECH, Ortenberg, Germany). Cell viability was expressed as MTT levels detected in stimulated cells versus untreated (MED) THP-1dM cells.

### 4.3. RNA Extraction and Real Time Quantitative PCR (qPCR)

Total RNA was extracted using the RNeasy Mini kit (Qiagen, Milan, Italy), according to the manufacturer’s instructions. RNA concentration was determined spectrophotometrically at 260 nm. RNA (2 µg) was retro transcribed into cDNA using the SuperScript VILO cDNA Synthesis Kit (Invitrogen, Waltham, MA, USA) under the following conditions: 10 min at 25 °C and 60 min at 42 °C. The reaction was terminated at 85 °C for 5 min, and cDNAs were stored at −20 °C. For each target, 50 ng cDNA from total RNA were amplified in triplicate in the ABI Prism 7500 HTSequence Detection System (Applied Biosystems). The 5× HOT FIREPol^®^ EvaGreen^®^ qPCR Mix Plus (ROX) (Solis BioDyne, Tartu, Estonia) was used under the following conditions: 95 °C for 15 min and 40 cycles of 95 °C for 15 s, 62.5 °C for 20 s, 72 °C for 20 s. For relative quantification of each target vs. β-actin mRNA, the comparative CT method was used. The sequences of the used primers (Sigma-Aldrich, St. Louis, MI, USA) are reported in Appendix A.

### 4.4. Protein Extraction and Western Blot Analysis

THP-1 dM cells were lysed using Cell Extraction Buffer (BioSource, Thermo Fisher Scientific, Waltham, MA, USA) containing 1 mM PMSF and protease and phosphatase inhibitor cocktail (Sigma-Aldrich) (1:200 and 1:100). After incubation for 30 min on ice, lysates were centrifuged at 12,000× *g* for 10 min at 4 °C and the pellets were discarded. Cytosol protein concentration was determined by Bradford assay at 595 nm. Cytosol proteins (20 μg) were separated by electrophoresis using 4–12% NuPAGE^®^ Bis-Tris gels (Life Technologies-Invitrogen, Carlsbad, CA, USA) and blotted on nitrocellulose membrane (GE Healthcare-Life Sci, Milan, Italy). Blots were blocked 1 h at room temperature on a shaker in 5% fat-free dried milk in TBS-T buffer (50 mM Tris-HCl pH 7.6, 200 mM NaCl, 0.1% Tween 20). Blots were incubated overnight on a shaker at 4 °C with the following antibodies (Abs): anti-phospho-ERK1/2 (1:300,Cell-Signaling, Danvers, MA, USA), anti-phospho-AKT (S473 1:500, Cell Signaling), anti-phospho-p38 (T180/Y182 1:350, Cell Signaling, Danvers, MA, USA), anti-phospho-p70S6K (1:500 Cell Signaling), anti-phospho-Tau (Ser262 1:500, Thermo Fisher Scientific, Waltham, MA, USA), anti-beclin-1 (1:1000, Cell Signaling), anti-LAMP2A (1:900, Abcam, Cambridge, UK), anti-LC3B (1:1000, Cell Signaling), anti-hsc70 (1:3000, Abcam), and anti-SQSTM1/p62 (1:1000, Cell Signaling). Abs were diluted in 5% fat-free dried milk in TBS-T buffer. A mouse anti-β-actin Ab (1:40,000, Sigma-Aldrich) was used as a standard to normalize the signals for each target protein. Preliminary experiments, using specific Abs against the total form of studied protein kinases, were performed, and no effect of GNVs was observed. After 1 h incubation with a peroxidase-linked anti-rabbit/mouse (1:6000 or 1:8000; Sigma-Aldrich) IgG secondary Ab, signals were measured by chemiluminescence reagents (ECL Plus Kit; Amersham, Little Chalfont, Amersham, UK), detected with a CCD camera using ImageQuant 800 system (Amersham), quantified using Image-J software 1.48v, and expressed as ratios between the target protein and β-actin.

### 4.5. Intracellular ASC Protein Staining and Image Stream Analysis by FlowSight AMNIS

THP-1dM cells treated as described above were permeabilized with 100 μL of Saponine in PBS (0.1%) (Life Science VWR, Lutterworth, Leicestershire, UK), and 5 μL (25 μg/mL) of the PE-anti human ASC (clone HASC-71, isotype mouse IgG1, Biolegend, San Diego, CA, USA) mAb was added to the tubes for 1 h at 4 °C. Cells were then washed with PBS and centrifuged at 1500× *g* for 10 min at 4 °C. Finally, the cells were fixed with 100 μL of PFA in PBS (1%) (BDH, UK) for 15 min, washed with PBS, centrifuged at 1500× *g* for 10 min at 4 °C, resuspended in 50 μL of iced PBS, and analyzed by AMNIS FlowSight. Results were analyzed by the IDEAS analysis software 6.2 (Cytek Biosciences, Fremont, CA, USA). The Apoptosis-associated speck-like-protein containing CARD speck formation was analyzed by internalization feature, utilizing a mask that represents the whole cell defined by the bright field (BF) image and an internal mask defined by eroding the whole cell mask, differentiating diffuse or spot (speck) fluorescence inside of cells. A threshold mask was used to separate all ASC positive cells population between ASC-speck spot cells or ASC-diffuse cells by the different diameter of the spot area. In ASC-speck, the spot shows a small area and high max pixel, and the opposite occurs in ASC-diffuse cells. The analysis of NLRP3 expression was performed by internalization feature utilizing a mask representing the whole cell, defined by the brightfield image, and an internal mask defined by eroding the whole cell mask.

### 4.6. Cytokine Production and Caspase-1 (p20) Release

Simple Plex Assays for IL-18 (SPCKB-PS-000501), IL-1β (SPCKB-PS-000216), and caspase-1 (p20 subunit) (SPCKB-PS-003613) were run by an automated immunoassay system (ELLA) (Biotech, Minneapolis, MN, USA), using a microfluidic cartridge that automates all steps of the immunoassay. Frozen supernatants of THP-1dM cells treated as described above were centrifuged to remove particulates and tested immediately according to manufacturer’s instruction. Each cartridge is composed of channels that contain glass nano reactors (GNRs), which are the core of a Simple Plex immunoassay. Each channel of the cartridge contains three GNRs coated with a capture antibody, so that each sample is automatically processed in triplicate. The limit of detection (LOD) of human IL-1β is 0.064 pg/mL, for IL-18 is 0.2 pg/mL, and for Caspase-1 is 0.04 pg/mL. The LOD was calculated by adding three standard deviations to the mean background signal determined from multiple runs.

### 4.7. Statistical Analysis

Experiments were repeated at least three times. Data were tested for normality by Shapiro–Wilk test. For data regarding ELLA and Flow cytometry AMNIS analysis, as not-normally distributed, statistical significance was calculated using Mann–Whitney U tests.

For Western blot and qPCR analyses, data are expressed as the mean ± S.E.M of at least 3 independent experiments. Repeated measures one-way ANOVA, followed by Tukey’s multiple comparisons post hoc test, were used to assess the significance of differences between groups. Results were considered to be statistically significant if they were below the *p* < 0.05 threshold. Statistical analysis was performed using the GraphPad prism 8.4.

## Figures and Tables

**Figure 1 pharmaceuticals-16-01725-f001:**
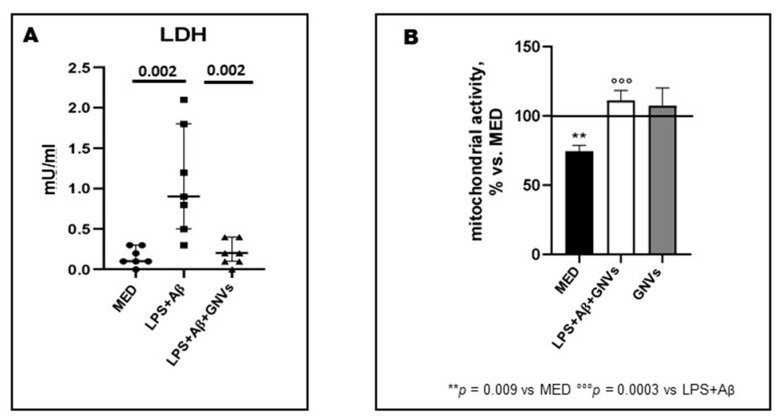
Effect of GNVs in LDH-release (**A**) and mitochondrial activity (**B**). THP-1 dM cells were treated with lipopolysaccharide (LPS) (1 µg/mL) for 2 h and with Aβ for 22 h with/without GNVs. (**A**) Cytotoxicity was assessed by quantifying lactate dehydrogenase (LDH) release. LDH activity in supernatants was measured as mU/mL by the rate of reduction of NAD+ to NADH after addition of the LDH reaction mix. Mitochondrial activity (**B**) assessed by MTT assay after GNVs treatment for 22 h (gray box) and after 2 h LPS priming followed by 22 h Aβ treatment in presence/absence of GNVs. Values are expressed as % vs. untreated (MED) THP-1dM cells. Significant differences were shown.

**Figure 2 pharmaceuticals-16-01725-f002:**
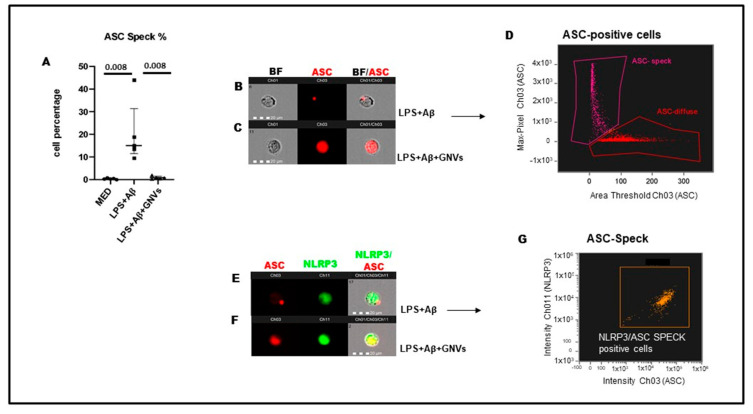
NLRP3 and ASC-speck formation in THP-1 dM cells were treated with lipopolysaccharide (LPS) (1 µg/mL) for 2 h and with Aβ for 22 h with/without GNVs. (**A**) The percentages of ASC-speck positive cells analyzed by FlowSight AMNIS and modulation of inflammasome effector proteins by Glibenclamide-loaded nanovectors (GNVs) in THP1 dM cell lines. Representative images of ASC-speck formation in LPS + Aβ stimulated cells: ASC-speck (**B**) and ASC-diffuse (**C**). The first column shows cells in brightfield (BF), second column shows ASC-PE (red) fluorescence, third column shows ASC-PE merged with BF (IDEA software 6.2). The percentage of ASC speck positive cells was performed using the same mask of internalization feature differentiating spot (speck) or diffuse fluorescence inside of cells (**D**); threshold mask was used to separate all ASC positive cells population in ASC-speck spot cells or ASC-diffuse cells by the different diameter of the spot area (**D**). In ASC-speck cell, the spot shows a small area and high max pixel; conversely, in ASC-diffuse cell, the fluorescence shows a large area and low max pixel representative images of NLRP3 and ASC-speck formation (**E**,**F**): The first column shows cells in ASC-PE fluorescence, second column shows NLRP3-FITC fluorescence, third column shows florescence of ASC merged with NLRP3 (IDEA software 6.2). The percentage of double positive cells (ASC speck and NLRP3 positive cells) was performed using the same mask of internalization feature (**G**) (IDEA software 6.2).

**Figure 3 pharmaceuticals-16-01725-f003:**
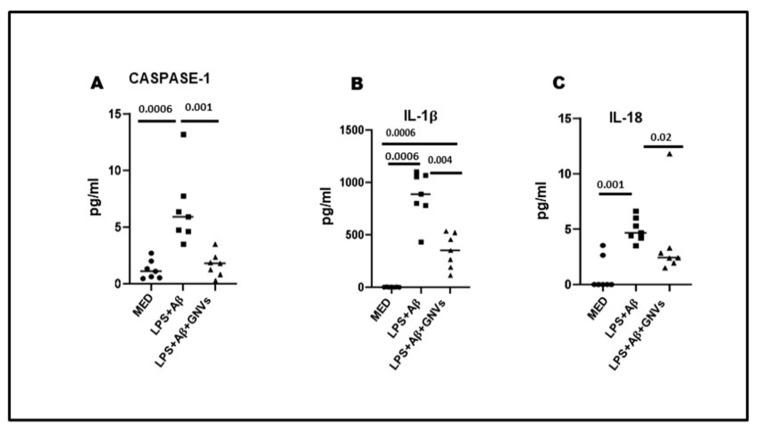
Effect of GNVs on NLRP3 activation; the supernatants of cell cultures were collected, and caspase-1 (**A**), IL-1β (**B**), and IL-18 (**C**) release was measured by Automated Immunoassay (ELLA). THP-1 dM cells were treated with lipopolysaccharide (LPS) (1 µg/mL) for 2 h and with Aβ for 22 h with/without GNVs. Data are representative of three independent experiments and expressed as mean ±  SD. Significant differences are shown.

**Figure 4 pharmaceuticals-16-01725-f004:**
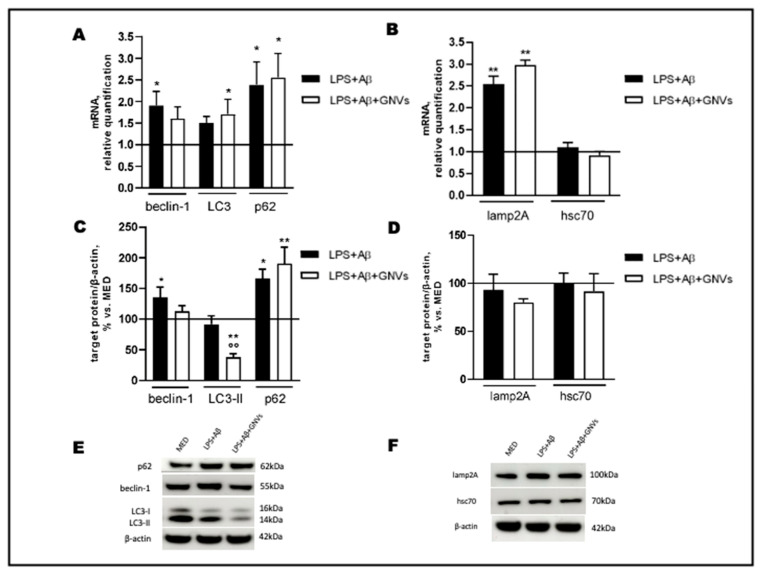
Effect of GNVs on autophagy markers. THP-1 dM cells were treated with lipopolysaccharide (LPS) (1 µg/mL) for 2 h and with Aβ for 22 h with/without GNVs. Relative quantification, calculated as ratio to β-actin, of mRNA levels (**A**,**B**) of macroautophagy (beclin-1, LC3, and p62) (**A**) and CMA (lamp2A, hsc70) (**B**) markers. Protein expression (**C**,**D**), normalized to β-actin, of macroautophagy (beclin-1, LC3-II and p62) (**C**) and CMA (lamp2A, hsc70) (**D**) markers. Representative Western blot images (**E**,**F**), showing immunoreactivity for the target proteins and β-actin, were used as an internal standard. Repeated measures one-way ANOVA, followed by Tukey’s multiple comparisons post hoc test; * *p* < 0.05, ** *p* < 0.01 vs. untreated cells (MED, 100%); °° *p* < 0.01 vs. LPS + Aβ.

**Figure 5 pharmaceuticals-16-01725-f005:**
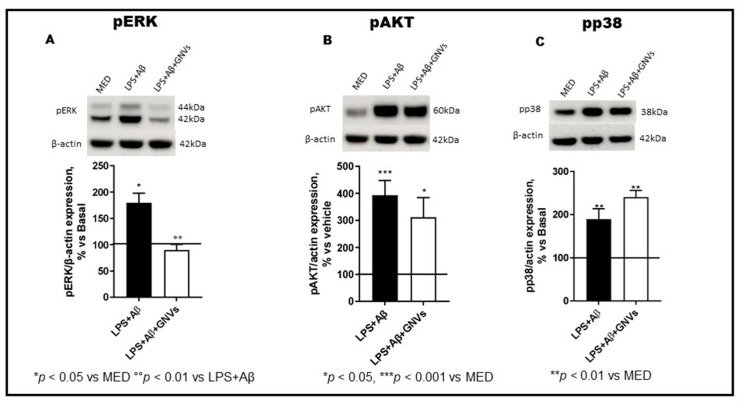
Effect of GNVs on ERK1/2, AKT, and p38 phosphorylation. The phosphorylation status of (**A**) ERK1/2, (**B**) AKT, and (**C**) p38 were investigated in THP-1 dM cells treated with lipopolysaccharide (LPS) (1 µg/mL) for 2 h and with Aβ for 22 h with/without GNVs by Western blot analysis (WB). Each result was expressed as the percentage of ratios between phosphorylated kinases and the corresponding β-actin expression versus un-treated THP-1 dM cells (MED, 100%). Repeated measures one-way ANOVA, followed by Tukey’s multiple comparisons post hoc test (*** *p* < 0.0001, ** *p* < 0.001, * *p* < 0.05 vs. MED; °° *p* < 0.01 vs. LPS + Aβ). Representative blots images are shown in the upper part of panels (**A**–**C**).

**Figure 6 pharmaceuticals-16-01725-f006:**
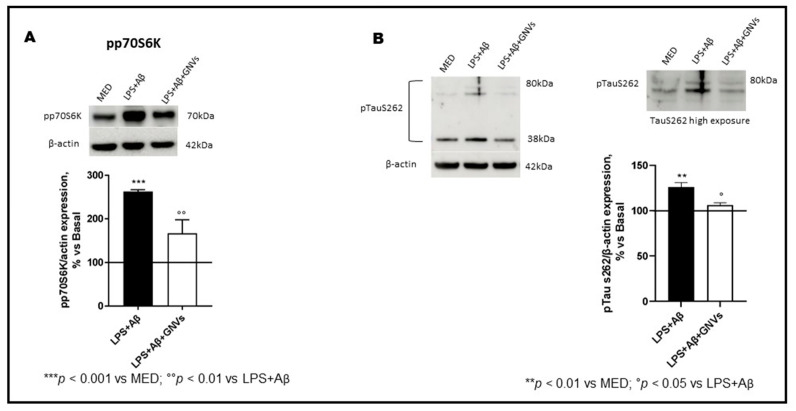
Effect of GNVs on p70S6K and Tau Ser262 phosphorylation. The phosphorylation status of (**A**) p70S6K and (**B**) TauS262 were investigated by Western blot analysis in THP-1 dM cells treated with lipopolysaccharide (LPS) (1 µg/mL) for 2 h and with Aβ for 22 h with/without GNVs. Each result was expressed as the percentage of ratios between phosphorylated proteins and the corresponding β-actin expression versus un-treated THP-1 dM cells (MED, 100%). Repeated measures one-way ANOVA, followed by Tukey’s multiple comparisons post hoc test (*** *p* < 0.0001, ** *p* < 0.001 vs. MED; ° *p* < 0.05, °° *p* < 0.01 vs. LPS + Aβ). Representative blots images are shown in the upper part of panels (**A**,**B**).

## Data Availability

Data is contained within the article or Appendix A.

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
