# Peer review of "Glibenclamide-Loaded Engineered Nanovectors (GNVs) Modulate Autophagy and NLRP3-Inflammasome Activation"

_pharmaceuticals, 2023, doi:10.3390/ph16121725_

Round 1

Reviewer 1 Report

Comments and Suggestions for Authors

In the manuscript “Glibenclamide-Loaded Engineered Nanovectors (GNVs) modulate autophagy and NLRP3-Inflammasome activation” Sarasella et al. analyze the effect of GNV on the NLRP3 inflammasome activation and autophagy. The manuscript is well structured with clear writing style and experimental settings, and the data are all well presented with. However, there are some concerns which should be reflected.

1.       A major concern is related with the experiments used to demonstrate changes in the autophagy in the analyzed cells which are not completely convincing.

·         Accepted methods for detection of autophagy do not include p62, LC3 and beclin-1 gene expression level. Rather the autophagic flux which is measured (using WB) by: 1. Conversion of LC3I to LC3II, in the presence of autophagy inhibitors such as bafilomycin-A1 or chloroquine. 2. Reduction of the protein level of p62 which is directly bind to LC3 and degraded by autophagy (Pankiv et al., 2007, Yoshii et al., 2017, Klionsky et al, 2020)

·         The authors should explain the changes in the LC3i to II conversion between the untreated control and the LPS+Aβ sample.

2.       Figure 5A, B, C Actin blots are saturated. Since the authors do not use the total ERK, AKT and p38, respectively as a control for the phosphorylated proteins, they should show lower exposure of the actin WBs.

Minor points:

1.       Introduce the abbreviation Aβ.

2.       In figure 1A: LPS +Aβ (not aβ)

3.       Line 112: Aβ42  or Aβ in the whole text.

4.       Line 114: 24 hours not shown? Which time point is shown?

5.       Please correct the descriptions on figure 2 (βAMY); 2B (aβ). Use an uniform abbreviation of the Aβ throughout the paper including the figure legends and figure terms.

6.       GNV can be used through the paper including the figure legends, once introduced as an abbreviation.

7.       Include the size (in KD) of the detected proteins.

Author Response

pharmaceuticals-2739617

REVIEWER 1

In the manuscript “Glibenclamide-Loaded Engineered Nanovectors (GNVs) modulate autophagy and NLRP3-Inflammasome activation” Sarasella et al. analyze the effect of GNV on the NLRP3 inflammasome activation and autophagy. The manuscript is well structured with clear writing style and experimental settings, and the data are all well presented with. However, there are some concerns which should be reflected.

Q1:  A major concern is related with the experiments used to demonstrate changes in the autophagy in the    analyzed cells which are not completely convincingAccepted methods for detection of autophagy do not include p62, LC3 and beclin-1 gene expression level. Rather the autophagic flux which is measured (using WB) by: 1. Conversion of LC3I to LC3II, in the presence of autophagy inhibitors such as bafilomycin-A1 or chloroquine. 2. Reduction of the protein level of p62 which is directly bind to LC3 and degraded by autophagy (Pankiv et al., 2007, Yoshii et al., 2017, Klionsky et al, 2020). The authors should explain the changes in the LC3i to II conversion between the untreated control and the LPS+Aβ sample

A1: We fully agree with the Reviewer that accepted methods for autophagic flux measure include WB analyses to assess the conversion of LC3I to LC3II and the expression of the autophagy substrate p62, both under basal conditions and in presence of an autophagy inhibitor (i.e. bafilomycin-A1 or chloroquine). As known, the conversion from LC3I to the lipidated form LC3II occurs upon autophagy induction and correlates with autophagosome accumulation. Anyway, a consensual LC3 mRNA increase has been often observed during autophagy induction (Nara et al, Cell Struct. Funct. 2002;27:29–37; Kanzawa et al, Cell Death Differ. 2004;11:448–457), and gene expression of autophagy markers can give additional information useful to better understand results on protein expression. And this is the case.

The Reviewer should consider that all WB analyses on the most common autophagy markers were done and that, in addition, results from qPCR were provided. Collectively, the observed decrease of LC3II protein expression in presence of GNVs, together with the induction of LC3 gene expression, is coherent with our conclusion of a potentiation of macroautophagy ascribable to GNVs.

Moreover, to answer to the concern at point 1, we performed a new experiment in cells exposed to the same stimuli (medium, LPS+Ab, LPS+Ab+GNVs) in presence of 100nM bafilomycin-A1 and we analyzed the protein expression of the autophagosome marker LC3II, confirming results obtained in absence of the inhibitor. As expected, bafilomycin-A1 increased LC3II expression in all analyzed samples with respect to the same samples in absence of the inhibitor, and a mild reduction of LC3II was confirmed in LPS+Ab-treated cells and, most importantly, a marked reduction of LC3II was confirmed in LPS+Ab+GNVs-treated cells.

      THP-1dM + bafilomycin-A1 (100nM/24h):

Uncropped images:

The Reviewer should also consider that we decided not to use bafilomycin-A1 in our first submission based on previous experience on autophagy studies suggesting that the use of this kind of inhibitor is mandatory in case of LC3II protein increase (and not decrease) to discriminate between an induction of the autophagic flux rather than an autophagosome accumulation due to a block of the final autophagy steps responsible for autophagosome degradation.

Concerning the issue at point 2, we clarify that it is not surprising that p62 was not reduced in LPS+Ab+GNVs-treated cells. As a matter of fact, as previously reported in a similar experimental setting (Lonati et al, Int J Mol Sci. 2023 May 24;24(11):9201), p62 mRNA and protein levels increase in response to inflammatory stimuli, represented in the cited paper by proinflammatory cytokines and in our study by LPS+Ab-treatment. These increases could be explained considering the multiple roles exerted by p62 in pathways activated in response to inflammatory stimuli; p62, in fact, is not only involved in autophagy degradation, but it is also a hub for pro-and anti-inflammatory pathways positively regulating transcription factors under stress conditions (Sánchez-Martín et al, FEBS J. 2019;286:8–23; Hennig et al, Biomedicines. 2021;9:707).  Since we observed that the exposure to GNVs was unable to markedly modify the increase in mRNA and protein levels of p62 induced by LPS+Ab, we could speculate that p62 might be mainly engaged in the above-cited regulatory pathways rather than only in the autophagic machinery.

We revised the paragraph “2.3 GNVs effect on macroautophagy and CMA” to address the Reviewer’s concerns.

Q2  Figure 5A, B, C Actin blots are saturated. Since the authors do not use the total ERK, AKT and p38, respectively as a control for the phosphorylated proteins, they should show lower exposure of the actin WBs.

A2: Thank you for the comment, we have replaced the saturated actin blots images with lower exposure in Fig. 5A pERK (ex Fig.4A), 5B pAKT (ex Fig. 4B) , 5C pp38 (ex Fig.4C) and 6A pp70S6K (ex Fig. 5A)

Minor points:

  1. Introduce the abbreviation Aβ.:

A1: Sorry, we introduce the abbreviation Aβ in the revised text.

  1. In figure 1A: LPS +Aβ (not aβ)

A2: Sorry for the mistake; it was resolved in the revised Figures.

  1. Line 112: Aβ42  or Aβ in the whole text.

A3: Yes definitely, we will correct with “Aβ” in the whole text

  1. Line 114: 24 hours not shown? Which time point is shown?

A4:  Sorry there was a mistake in the text, we have shown 22h hours of GNVs treatment following 2h LPS priming.

     We modified the  revised text as follows:

Line 117: “To evaluate the possible antioxidant role of GNVs, mitochondrial activity was analyzed next by MTT. Results showed that LPS+Aβ stimulation was harmful for mitochondria, reducing their activity (vs. medium alone: p < 0.01); this was completely prevented by GNVs (vs. LPS+Aβ p<0.001) (Figure 1B).GNVs treatment did not modify mitochondrial activity compared to untreated cells (MED), for this reason GNVs are biocompatible”.

  1. Please correct the descriptions on figure 2 (βAMY); 2B (aβ). Use an uniform abbreviation of the Aβ throughout the paper including the figure legends and figure terms.

A5: Thanks, we will correct the mistakes in the revised Figures and text.

  1. GNV can be used through the paper including the figure legends, once introduced as an abbreviation.

A6: Of course, we will fix the oversight in the correct text

  1. Include the size (in KD) of the detected proteins.

A7: As the Reviewer’s suggestion, we added molecular weights in KDa near to the detected proteins; then we deleted the molecular weight in the relative legends in Figure 5 and 6.

Reviewer 2 Report

Comments and Suggestions for Authors

Description:

In the present study, Saresella M et al., describes how autophagy induction, suppresses the beta-amyloid formation which is the causative agent for NLRP3 inflammasome formation during 

 Alzheimer’s Disease (AD). They claimed that liposome mediated nanovectors formulated with Glibenclamide enhanced the autophagy, responsible for clearance of beta amyloid during neurodegenerative diseases like AD. Due to beta amyloid generation the mitochondrial heath gets compromised and leads to production of proinflammatory cytokines and finally causes activation of NLRP3 inflammasome formation. I would advise authors to include suggested points with proper references to make it presentable and impactful. Will be happy to see the corrected version of manuscript before the final acceptance. 

Major Points: 

1.     In figure 1B: authors need to add some mitochondrial dye like CMSros or JC1 to check the mitochondrial heath apart from MTT assay. 

2.     How author claiming so assertively about the activation of only NLRP3 inflammasome is activated not AIM2 or NLRC4 in AD set up without showing any supportive data relevant to that. Even there is no data showed related to the expression of NLRP3 like western blot. This is basic experiments and would suggest including it. 

3.     It’s advised to include the simple fluorescence images of the cells with ASC specs or AMNIS flow cytometry with all different conditions.

4.     Author showed that induction of autophagy upon GNV treatment in Fig-3E. Could you explain why the LC3-II band seems more in medium control while upon beta-amyloid treatment it started decreasing. Possible reason may be due to activation of autophagy at the start degradation of the protein. But in this set up use the autophagy inhibitor like bafilomycin-A to prove autophagy is solely responsible for proteasomal degradation. It is not very clear from the p62 expression. Please include the lower exposer of p62, beclin-1 and Lamp2A. 

5.     Glibenclamide is an inhibitor of K+ efflux channel, directly suppressed the NLRP3 inflammasome complex. It’s advisable to include the western blots for the expression of NLRP3. 

Minor Points:

1.     There are mainly three types of autophagy, macroautophagy, microautophagy and chaperon mediated autophagy. Please describe it in the introduction part.  

2.     Keep the abbreviations same throughout the manuscript. e.g., in Fig-2: GENV instead of GNV, Ab instead of BAMY and aB etc.

Comments on the Quality of English Language

Mine editing is required. Rest are fine. 

Author Response

pharmaceuticals-2739617

REVIEWER 2

Comments and Suggestions for Authors

Description:

In the present study, Saresella M et al., describes how autophagy induction, suppresses the beta-amyloid formation which is the causative agent for NLRP3 inflammasome formation during 

 Alzheimer’s Disease (AD). They claimed that liposome mediated nanovectors formulated with Glibenclamide enhanced the autophagy, responsible for clearance of beta amyloid during neurodegenerative diseases like AD. Due to beta amyloid generation the mitochondrial heath gets compromised and leads to production of proinflammatory cytokines and finally causes activation of NLRP3 inflammasome formation. I would advise authors to include suggested points with proper references to make it presentable and impactful. Will be happy to see the corrected version of manuscript before the final acceptance. 

Major Points: 

Q1.     In figure 1B: authors need to add some mitochondrial dye like CMSros or JC1 to check the mitochondrial heath apart from MTT assay. 

A1 We thank the reviewers for the suggestion, but the experiment conducted with MTT ASSAY assesses mitochondrial function to indicate the energy capability, and  MTT values were normalized to the number of plated cells.

As is well known, if the mitochondria are functioning, they produce energy and prevent oxidative stress. In fig.1B we investigated mitochondria activity, we did not test cell apoptosis. In fact, JC-1 dye is widely used in apoptosis studies to monitor mitochondrial health, but this is not the aim of our experiment. We are sorry, but we do not know CMSros, if it is MitoTracker dye, we could test it for mitochondria staining in other next experiments, studying the cell cycle or processes such as apoptosis. In this paper we didn’t investigate THP-1dM apoptosis.

Q2.     How author claiming so assertively about the activation of only NLRP3 inflammasome is activated not AIM2 or NLRC4 in AD set up without showing any supportive data relevant to that. Even there is no data showed related to the expression of NLRP3 like western blot. This is basic experiments and would suggest including it. 

A2: Thanks for point out this question; the NLRP3 inflammasome is crucial in the neuroinflammatory pathway and has recently been highlighted as a potential target for AD treatment. Among the many reported inflammasomes, the NLRP3 is currently the most studied one. Just like the structure of the above-mentioned inflammasomes, the NLRP3 inflammasome includes the sensor protein NLRP3, the adaptor protein apoptosis-associated speck-like protein containing a CARD (caspase activation and recruitment domain) (ASC), and the effector protein (pro-caspase-1, a cysteine protease) (Schroder and Tschopp 2010). These three proteins can interact closely to regulate the function of NLRP3 inflammasome. Once NLRP3 recognizes the foreign pathogen molecules or internal danger signals, it will be activated and undergos self-oligomerization. Then NLRP3 binds to the pyrin domain (PYD) domain of the adaptor protein ASC, and recruits the protease pro-caspase-1 to form the NLRP3 inflammasome, which cleaves pro-caspase-1 into activated caspase-1 through autocatalysis. The activated caspase-1, as an inflammasome effector protein, is able to cleave the inactive pro-inflammatory cytokines pro-IL-1β and pro-IL-18 into mature forms of IL-1β and IL-18, respectively. Ultimately, IL-1β and IL-18 are released outside of the cell to play a variety of non-specific inflammatory roles (Martinon et al., 2002; Kelley et al., 2019). In addition, the activated caspase-1 can also mediate a type of inflammatory-related programmed cell death, which is called pyroptosis. A large amount of inflammatory substances released after cell pyroptosis will induce a strong inflammatory response (Fink and Cookson 2006; Shi et al., 2015).More and more experimental evidence show that the activation of NLRP3 inflammasome is closely related to neurodegenerative diseases (Duan et al., 2020; Feng et al., 2021). Under the stimulation of Aβ plaques and tau aggregates, microglia and astrocytes mediate chronic neuroinflammatory response, neuronal death and pyroptosis through intracellular NLRP3 inflammasome, thereby driving the occurrence and progression of AD (Han et al., 2020b; Van Zeller et al., 2021). More importantly, pharmacological inhibition of NLRP3 inflammasome exhibits neuroprotective effects. The use of inhibitory treatment against NLRP3 inflammasome can reduce Aβ deposition and alleviate the cognitive impairment of AD mice (Yan et al., 2020b).  Actually, only NLRP3 inflammasome  was related to the Authophagic activity. It is important to understand this crosstalk between inflammation and autophagy, as it applies to various inflammatory diseases. The current available information about this mutual regulation that exists between NLRP3 inflammasome and autophagy is not already understood. In our previous study we started to investigate NLRP3 and Autophagy (La Rosa et al., 2020) and in our opinion  is an important point of attention.

At present, in vitro experimental protocols examining inflammasome activation commonly include “priming” with a TLRs agonist, such as lipopolysaccharide (LPS), and a second stimulus, such as Aβ or different canonical NLRP3 stimuli including nigericin, ATP, and urea crystals. THP-1 cell lines constitutively express NLRP3 (Guzova et al., 2019); thus, NLRP3 colocalization with ASC, ASC Speck formation and related cytokines production are the important things to evaluate NLRP3 activation or not in response to different drug treatments.

To better underline these points in the revised text,  we  have included a new panels (Figure 2 B,C,D,E,F,G) which reported the ASC-speck activation and ASC-speck/NLRP3 colocalization in THP-1 at the different experimental conditions; the image was obtained by AMNIS Flowsight.  We are very sorry not to be able to satisfy the Reviewer request for WB analysis,  but at the moment we don’t have the necessary materials.  The text was modified as follows:

Line 87 “In recent years, a large amount of data from cell experiments and animal models have confirmed that the activation of NLRP3 inflammasome can also affect the deposition and spread of Aβ. APP/PS1 mice, compared to  NLRP3 and caspase-1 knockout AD model mice have a significantly enhanced ability of microglia to phagocytose Aβ and differentiate microglia into anti-inflammatory M2 type, which facilitates Aβ clearance [13].”

Line: 293: “Notably, THP-1 cell lines constitutively express NLRP3 [48]; thus, to better investigate NLRP3 activation or not in response to GNVs treatments we evaluate  NLRP3 colocalization with ASC, ASC Speck formation and related cytokines production.”

Q3.     It’s advised to include the simple fluorescence images of the cells with ASC specs or AMNIS flow cytometry with all different conditions.

A3. We Thank the Reviewer for this tip; as you suggested we have included, in the revised text, the new panel in the Figure 2(panel B).

Q4.     Author showed that induction of autophagy upon GNV treatment in Fig-3E. Could you explain why the LC3-II band seems more in medium control while upon beta-amyloid treatment it started decreasing. Possible reason may be due to activation of autophagy at the start degradation of the protein. But in this set up use the autophagy inhibitor like bafilomycin-A to prove autophagy is solely responsible for proteasomal degradation. It is not very clear from the p62 expression. Please include the lower exposer of p62, beclin-1 and Lamp2A. 

A4: We thank the Reviewer for pointing out this question, which allows us to better clarify and discuss the results on LC3-II protein expression obtained in LPS+Ab-treated cells and shown in Figure 4E. As remarked by the Reviewer, a trend to decreased LC3-II levels emerged also in cells exposed to LPS+Ab as compared to medium-treated cells, and this could be interpreted as an initial protective cell response against the inflammatory/toxic stimuli. Anyway, as shown in the histogram (Figure 4C), only a mild and not significant reduction of LC3-II was obtained from the analysis of 3 independent experiments, which did not allow us to conclude that exposure to LPS+Ab was sufficient to significantly reduce LC3-II. Moreover and supporting this view, as shown in Figure 4A, a trend to increased LC3 mRNA levels was also found in LPS+Ab-treated cells, which reached statistical significance only after exposure to GNVs. Collectively, these results suggest that a mild autophagy induction is ongoing in cells after LPS+Ab exposure, and that autophagy markedly increases following GNVs treatment.

To further verify the above described hypothesis, we performed a new experiment in cells exposed to the same stimuli (medium, LPS+Ab, LPS+Ab+GNVs) in presence of 100nM bafilomycin-A1 and we analyzed the protein expression of the autophagosome marker LC3II, confirming results obtained in absence of the inhibitor. As expected, bafilomycin-A1 increased LC3II expression in all analyzed samples with respect to the same samples in absence of the inhibitor, and a mild reduction of LC3II was confirmed in LPS+Ab- treated cells and, most importantly, a marked reduction of LC3II was confirmed in LPS+Ab+GNVs-treated cells.

THP-1dM + bafilomycin-A1 (100nM/24h):

Uncropped images:

In response to the Reviewer's concern on p62 results,  we clarify that it is not surprising that p62 was not reduced in LPS+Ab+GNVs-treated cells. As a matter of fact, as previously reported in a similar experimental setting (Lonati et al., 2023), p62 mRNA and protein levels increase in response to inflammatory stimuli, represented in the cited paper by proinflammatory cytokines and in our study by LPS+Ab treatment. These increases could be explained considering the multiple roles exerted by p62 in pathways activated in response to inflammatory stimuli; p62, in fact, is not only involved in autophagy degradation, but it is also a hub for pro-and anti-inflammatory pathways positively regulating transcription factors under stress conditions (Sánchez-Martín et al., 2021).  Since we observed that the exposure to GNVs was unable to markedly modify the increase in mRNA and protein levels of p62 induced by LPS+Ab, we could speculate that p62 might be mainly engaged in the above-cited regulatory pathways rather than only in the autophagic machinery.

Finally, we are sorry but we are unable to include lower exposures of p62, beclin-1 and Lamp2A. The Reviewer should consider that immunoreactive bands for all investigated targets are not saturated, thus implying that the obtained signals were considered suitable for the aim of the experiment and no lower exposure were made.

We revised the paragraph “2.3 GNVs effect on macroautophagy and CMA” to address the Reviewer’s concerns.

Q5.     Glibenclamide is an inhibitor of K+ efflux channel, directly suppressed the NLRP3 inflammasome complex. It’s advisable to include the western blots for the expression of NLRP3.

A5: As reported in the A2, the activation of NLRP3 was evaluated by ASC-speck formation, downstream protein IL-18, IL-1b and caspase -1 production. In the revised text we are including the news figures (2A-G) to better underline the ASC-speck/NLRP3 colocalization  in LPS+Ab stimulated cells in absence of GNVs.

The text was modified as follows:

Line 136: “The percentage of positive cells for ASC-speck formation (differently to ASC-diffuse) was selected (Figure 2D); the NLRP3 production and ASC-speck colocalization were investigated next by Flowsight AMNIS analyses in LPS+Αβ-stimulated THP1dM cells in the absence of GNVs (Figures 2E,F). Results were shown as percentage (median:11%) of double positive cells for ASC-speck and NLRP3 in LPS+Αβ-stimulated THP1dM in the absence of GNVs (Figure 2G). Addition of GNVs suppressed ASC-speck, thus the percentage of double positive cells was naught. Representative images are provided in Figures 2B,C, E,F.

Minor Points:

Q6.     There are mainly three types of autophagy, macroautophagy, microautophagy and chaperon mediated autophagy. Please describe it in the introduction part.  

A6: thanks for the comment, as you suggest, we have revised the text as follow:

Line 47: “Autophagy is a complex process that can be differentiated into three primary types of autophagy: microautophagy, macroautophagy, and chaperone-mediated autophagy (CMA). While each is morphologically distinct, all three culminate in the delivery of cargo to the lysosome for degradation and recycling[1].”

Q7.     Keep the abbreviations same throughout the manuscript. e.g., in Fig-2: GENV instead of GNV, Ab instead of BAMY and aB etc.

 A7: Sorry for the mistakes. Abbreviations and figures were revised.
